# A novel few-shot object detection framework for multi-scene driving based on contrastive proposal encoding

Yalei Dong[1], Jing Xiao[1,2], Fengchen Wei 🄳[3]*

1 Hebei Chemical and Pharmaceutical College, Shijiazhuang, China, 2 Tianjin University, School of Electrical and Information Engineering, Tianjin, China, 3 University of Sussex, Department of Engineering and Design, Brighton, United Kingdom

* fw216@sussex.ac.uk

## Abstract

This paper proposes a few-shot object detection algorithm based on FSCE, tailored for multi-scenario driving environments. Unlike existing methods that focus on single scenarios, our approach addresses challenges of cross-scenario heterogeneity and overfitting in low-data regimes. We enhance feature representation through a multi-scale feature module that integrates local and contextual information, and replace the traditional Softmax with a cosine Softmax classifier to reduce intra-class variance via L2 normalization and angular margin constraints. This work is the first to apply few-shot detection to both nighttime infrared and daytime visible-light driving scenarios. Experiments on FLIR and BDD100K demonstrate superior generalization and accuracy over existing methods. Future work will explore reducing model complexity while maintaining performance.

## Introduction

With the rapid advancement of autonomous driving technology and the increasing complexity and diversity of driving environments, object detection algorithms face significant challenges in ensuring accuracy, robustness, and real-time performance [1–5]. In typical daytime conditions, visible light imaging remains the primary sensing modality, due to its ability to capture detailed texture and structural features of objects. However, under low-light or nighttime conditions, visible light imaging is often limited by poor visibility and reduced contrast. In contrast, infrared imaging offers enhanced perception by capturing thermal radiation, allowing for reliable object detection even in challenging lighting conditions. Despite the advantages of infrared, the performance of target detection algorithms in both visible light and infrared domains heavily relies on large-scale annotated datasets, which remain scarce in practical applications due to high data collection costs, privacy concerns, and the difficulties inherent in capturing diverse real-world driving scenarios.

**Data availability statement:** The data underlying the results presented in the study are available from FLIR (https://oem.flir.com/solutions/automotive/adas-dataset-form/) and BDD100K (https://bair.berkeley.edu/blog/2018/05/30/bdd/).

**Funding:** The author(s) received no specific funding for this work.

**Competing interests:** The authors have declared that no competing interests exist.

Current research predominantly focuses on optimizing target detection algorithms for single-modality or single-scenario tasks. For instance, deep learning-based methods have been developed to improve detection accuracy in nighttime infrared images [1–3], while others focus on robust detection in daytime visible light conditions under variable lighting [4,5]. Although these approaches show promise within specific contexts, they face two critical challenges: (1) the heterogeneous nature of cross-modal data (e.g., infrared vs. visible light) complicates feature fusion and representation learning, and (2) conventional deep learning models are prone to overfitting when trained on small sample sizes, resulting in poor generalization to unseen scenarios.

To address these challenges, this paper introduces a novel few-shot target detection algorithm based on FSCE (Few-Shot Object Detection via Contrastive Proposal Encoding), designed to operate effectively across both visible light and infrared modalities in multi-scenario driving environments. Our approach focuses on improving knowledge transfer and model robustness under data-scarce conditions, with the goal of enabling accurate target detection in diverse and complex driving contexts.

This work contributes two key innovations: (1) The development of a multi-scale feature enhancement module, which incorporates local and contextual information from multiple scales, thereby improving the learning of few-shot classes. (2) The adoption of a cosine Softmax classifier with L2 normalization and angular margin constraints, reducing intra-class variance and improving classification accuracy under few-shot settings.

Experimental results on standard benchmarks such as FLIR and BDD100K demonstrate that the proposed method outperforms existing few-shot detection algorithms, showing superior generalization and robustness across both visible light and infrared driving environments.

## Related works

Object detection constitutes a fundamental task within the domain of computer vision, aimed at identifying and localizing one or more objects of interest within an image by annotating them with axis-aligned bounding boxes and providing a confidence score for each detection. The Faster R-CNN algorithm, introduced by Girshick et al. in 2016, exemplifies a classic two-stage object detection framework [6]. In the initial stage, it generates dense candidate regions utilizing a region proposal network (RPN). Subsequently, in the second stage, the algorithm applies a Region of Interest (RoI) pooling operation on these candidate regions, followed by category classification and bounding box regression, ultimately yielding the detection results.

Faster R-CNN is characterized by its two-stage approach, which first produces a substantial number of candidate regions before conducting detailed classification and regression for each region. While this sequential optimization strategy facilitates high detection accuracy, it is hindered by relatively slow inference speeds due to the processing of redundant candidate regions, rendering it less suitable for applications requiring high real-time performance. In contrast, single-stage detection algorithms adopt a more efficient paradigm: they predefine anchor boxes of varying scales

and aspect ratios on the image as reference points, allowing for simultaneous prediction of target categories and bounding box offsets through a single forward propagation. This design eliminates the need for an independent candidate region generation module, such as the RPN, thereby circumventing complex region proposal operations, significantly enhancing detection speed, and making it more appropriate for real-time applications. Notable examples of classic single-stage networks include the YOLO series (You Only Look Once) [7–12], the SSD (Single Shot Multibox Detector) [13], and RetinaNet [14].

In the domain of small-sample object detection, the dataset is categorized into two distinct subsets: a base class dataset and a new class dataset. The base class dataset encompasses a wide array of base classes, characterized by a substantial number of annotated images, thereby providing a rich foundation of data for model training. Conversely, the new class dataset comprises novel classes with a markedly limited number of annotated images, presenting significant challenges for model learning. It is important to emphasize that there is no overlap between the base class and the new class; they represent entirely independent categories of objects. The primary objective of small-sample object detection methodologies is to develop an efficient model capable of leveraging the extensive features learned from the base class dataset while also accommodating the limited features associated with the new class dataset. Ultimately, the model must exhibit robust generalization capabilities, enabling it to accurately detect the presence of both base class and new class objects within any given test image. This entails the model's ability to identify and detect both prevalent base class objects and rare new class objects, thereby fulfilling a critical role in complex and dynamic real-world application scenarios. Currently, the majority of small-sample image object detection approaches are predicated on the established two-stage object detection framework, Faster R-CNN. For instance, reference [15] employs Faster R-CNN as the foundational network architecture.

TFA [16] was proposed by scholars such as Xin Wang and Thomas E. Huang in 2020. Focusing on the task of feal-shot object detection, a two-stage fine-tuning method (TFA) was proposed: First, train the full model of Faster R-CNN on the basic classes of the PASCAL VOC, COCO, and LVIS datasets (with sufficient data). Then, fix the feature extractor to fine-tune only the last layer of the box predictor, and introduce the cosine similarity classifier to optimize the performance in low-sample scenarios. This method outperforms meta-learning methods such as FSRW and Meta R-CNN, as well as traditional fine-tuning methods on all three types of datasets. For example, the AP50 of the new class in the PASCAL VOC 1-shot scene reaches 25.3, and the AP of the rare class in LVIS increases by 13.3 percentage points.Bo Sun and other scholars published a paper in CVPR in 2021, proposing the few-shot object detection method FSCE [17]: Based on Faster R-CNN, through two-stage training (first training the model on the basic class data, and then fine-tuning on the balanced dataset of "basic class + new class", thawing the RPN and RoI feature extractors and optimizing the sample selection strategy), the contrastive branch and contrastive proposal coding (CPE) loss is introduced. Enhance instance-level intra-class compactness and inter-class differences to address the core issue that new classes are often mistakenly classified as similar base classes. This method achieves SOTA on both the PASCAL VOC and COCO datasets. In some scenarios of PASCAL VOC, nAP50 has improved by 8.8% compared to before, and COCO AP75 has improved by 2.7%. Moreover, it only requires regular batch training, and its memory efficiency is superior to the meta-learning method. Compared with the additional computational cost of branches, it can be ignored. The training process is compatible with the standard Faster R-CNN. Xiaopeng Yan et al. published in ICCV in 2019 and proposed the Meta R-CNN [18] model for solving instation-level few-shot learning tasks (such as object detection and segmentation). The model was evaluated on the PASCAL VOC and MS COCO datasets by introducing the Predictor-head Remodeling Network (PRN) on the basis of Faster/Mask R-CNN. Realize the meta-learning of the Region of Interest (RoI) features. PRN infers the category attention vector and implements a channel-level soft attention mechanism for RoI features, thereby dynamically adjusting the prediction head to adapt to new categories. The best model achieved a 51.5% mAP (10-shot) on VOC and significantly outperformed the baseline method on COCO. The inference time was only increased by approximately 1-2 ms per image compared to Faster R-CNN, demonstrating high efficiency and universality. This method can significantly improve the detection and segmentation performance with a small number of samples without the need for complex structures.Tang

Yingwei et al. proposed MSA-Net [19] in 2023, a few-shot object detection model based on a multi-stage attention mechanism. This model was evaluated on the PASCAL VOC and MS COCO datasets. Based on the improved Faster R-CNN, the gradient backpropagation decoupling mechanism, the attention distillation module (ABD), and the multi-scale attention module were introduced, effectively alleviating the optimization conflict between RPN and RCNN and enhancing the feature extraction and generalization capabilities. The best model achieved a 51.7% mAP (AP50) on PASCAL VOC (k=10) and a 27.6% mAP (AP50) on COCO (k=30), significantly outperforming baseline methods such as Meta R-CNN. Although the model does not explicitly report the inference time, it maintains a high computational efficiency through structural optimization and is suitable for real-time detection requirements in scenarios with few samples.Prannay Kaul, Weidi Xie and Andrew Zisserman proposed the pseudo-labeling method in 2022: Based on the Faster R-CNN model, the baseline detector is first constructed through the improved two-stage training. Then, the candidate detection results of new classes are obtained from the unlabeled images [20]. The kNN classifier is constructed using the self-supervised DINO ViT model to verify the candidate labels and eliminate false posipositive ones. The bounding box is corrected through the three-level class-independent regressiver. Ultimately, end-to-end re-training is carried out by combining high-quality pseudo-annotations with real annotations of basic classes. This method performs well on the PASCAL VOC and MS-COCO datasets. The nAP of the new class in the COCO 30-shot scene reaches 25.5 (an increase of 8.9 percentage points compared to the baseline), and the nAP50 of most scenes in PASCAL VOC ranks SOTA or second. And it can maintain the performance of the basic class (the bAP of the COCO basic class reaches 33.3).

Two-stage object detectors often suffer from overfitting to base classes during training, which compromises their ability to generalize to novel classes. To address this limitation, researchers have introduced meta-learning frameworks that incorporate meta-learners to enhance the generalization capacity of the feature extraction layers. These frameworks enable models to (1) rapidly adapt to novel class distributions, (2) improve learning efficiency under limited data conditions, and (3) effectively mitigate overfitting to base classes, thereby enhancing detection performance on novel categories. In 2019, Kun Fu et al. proposed a feth-shot object detection framework Meta-SSD [21] based on meta-learning, aiming at the problems of strong data dependence, time-consuming training and performance degradation in feth-shot scenarios of traditional object detection models. Based on the single-stage detector SSD and combined with the meta-learner to form a dual-component architecture, the meta-learner guides the detector to quickly adapt to new tasks with only one parameter update by learning the learnable learning rate of each parameter. Meanwhile, the NIST-FSD benchmark is constructed based on the Pascal VOC dataset. Experiments show that Meta-SSD performs well on the NIST-FSD benchmark. In the 5-way 1-shot scenario, its detection performance for unseen categories is superior to that of traditional SSD and the two-stage transfer learning method LSTD, and its performance tends to stabilize as the number of samples increases. It provides an important baseline for few-shot object detection driven by meta-learning. Bingyi Kang et al. proposed a fest-shot object detection model based on Feature Reweighting [22] in 2018. This research is one of the early explorations in the field of fest-shot object detection. The model was evaluated on the PASCAL VOC and MS COCO datasets. Based on the YOLOv2 detection framework, a meta feature learner and a lightweight reweighting module were introduced. Generate category-specific feature weight vectors by supporting set samples to dynamically adjust the feature response of the query image. The best model achieved a mAP of 47.2% on PASCAL VOC (10-shot) and a map of 9.1% on COCO (30-shot), significantly outperforming baseline methods such as OLOo-Joint and OLOO-FT. The model adopts a two-stage training strategy (base class training + few-shot fine-tuning) and designs a softmax classification loss to suppress redundant detection. Although the inference time was not explicitly reported, the model only requires approximately 1,200 iterations during the fine-tuning stage to converge, which is far less than the 25,000 iterations of the baseline method, demonstrating efficient learning and adaptability.Gongjie Zhang et al. proposed Meta-DETR [23] in 2022, a few-shot object detection model based on the image level. This model adopts Deformable DETR as the backbone network and introduces the Inter-Class Correlational Meta-Learning strategy. Multiple supporting classes are processed simultaneously through the Correlational Aggregation Module (CAM), effectively utilizing the correlations among classes to enhance the discrimination ability and generalization performance for similar classes. The model was evaluated on the

Pascal VOC and MS COCO datasets and significantly outperformed existing methods under different Settings ranging from 1-shot to 30-shot. For example, the average mAP was increased by 4.6% on VOC, and the performance was particularly outstanding on COCO. Meta-DETR does not require region proposals, avoiding the performance degradation caused by low-quality proposals in traditional methods. It has high inference efficiency and can scale at least the sample instance segmentation task.Limeng Qiao et al. proposed DeFRCN [24] (Decoupled Faster R-CNN) in 2021, a simple and efficient framework for few-shot object detection. This model is based on Faster R-CNN, introduces the Gradient decoupling layer (GDL) to achieve decoupling of multiple stages (RPN and RCNN) and multiple tasks (classification and localization), and uses the Prototype calibration module (PCB) for offline fraction calibration, which can improve the classification performance without additional training. On the PASCAL VOC and MS COCO datasets, DeFRCN significantly outperforms existing methods under different Settings from 1-shot to 30-shot. For example, the maps of 10-shot and 30-shot on COCO reach 18.5% and 22.6% respectively, which is approximately 6-8% higher than the previous best method. This model does not require complex meta-learning strategies, has high training and reasoning efficiency, and possesses excellent generalization ability and practicality.

At present, research on few-sample object detection mostly focuses on general scenarios, and there is a significant gap in the field of multi-scenario vehicle detection: First, the existing methods (such as TFA and FSCE) are mostly verified based on general datasets like PASCAL VOC, and have not been optimized for the illumination changes and occlusion differences in multiple scenarios (such as rainy days, nights, and tunnels) in vehicle detection, resulting in insufficient feature generalization ability. Second, models such as Meta-RCNN and DeFRCN do not take into account the few-sample distinction requirements for vehicle category segmentation (such as sedans, trucks, and new energy vehicles), and are prone to confusing similar vehicle models. Thirdly, although MSA-Net and others have introduced multi-scale attention, they have not designed an adaptation mechanism in combination with the target scale fluctuations in different scenarios in vehicle detection (such as being smaller when far away and larger when near). In response to this, this study focuses on multi-scenario vehicle few-sample detection, aiming to address issues such as scene adaptability, fine classification of vehicle models, and scale fluctuations, and to enhance vehicle detection performance in complex scenarios.

## Methodology

### 0.1 Basic network architecture

FSCE (Few-Shot Object Detection via Contrastive Proposal Encoding) [17], developed by researchers from the Institute of Automation, Chinese Academy of Sciences, and Tsinghua University, represents a significant advancement in the field of few-shot object detection through the incorporation of contrastive learning. As the first method to introduce contrastive learning into this domain, FSCE enhances the discriminative power of feature representations, thereby markedly improving detection performance under limited data conditions.

The core idea underlying FSCE is to optimize the feature representation of object proposals by leveraging contrastive learning to enforce semantic consistency. Specifically, it encourages the feature embeddings of proposals from the same class to be drawn closer together, while simultaneously pushing apart those from different classes. This approach directly addresses the data scarcity challenge inherent in few-shot learning, resulting in improved generalization to novel categories.

FSCE is implemented within the Faster R-CNN framework and introduces several key innovations: (1) the addition of a contrastive learning branch to the Region of Interest (RoI) head, which constructs a similarity measurement space for object proposals; (2) the design of the Contrastive Proposal Encoding (CPE) loss, which jointly optimizes intra-class compactness and inter-class separability; and (3) the preservation of an end-to-end training paradigm, allowing seamless integration without altering the fundamental training pipeline. As a result, the model achieves greater feature cohesion for instances of the same class while maintaining clear boundaries between different classes.

Our CPE loss is defined as shown in Formula 1, with considerations customized for detection. Specifically, for a mini-batch consisting of N RoI box features $\{z_i, u_i, y_i\}_{i=1}^{N}$, where $z_i$ represents the contrastive head encoded RoI feature for the *i-th* region proposal, $u_i$ indicates its Intersection-over-Union (IOU) score with the matched ground truth bounding box, and $y_i$ denotes the label of the ground truth.

$$L_{CPE} = \frac{1}{N} \sum_{i=1}^{N} f(u_i) \cdot L_{z_i} \tag{1}$$

$$L_{z_i} = \frac{-1}{N_{y_i} - 1} \sum_{j=1, j \neq i}^{N} \prod \{y_i = y_i\} \cdot \log \frac{\exp(\tilde{z}_i \cdot \tilde{z}_j / \tau)}{\sum_{K=1}^{N} \prod_{k \neq i} \cdot \exp(\tilde{z}_i \cdot \tilde{z}_k / \tau)} \tag{2}$$

Where $N_{y_i}$ is the count of proposals having the same label as $y_i$, and t is the hyper-parameter temperature, like that in InfoNCE. In Formula 2, $\tilde{z}_i = \frac{z_i}{\|z_i\|}$ stands for normalized features. Thus, $\tilde{z}_i \cdot \tilde{z}_j$ gauges the cosine similarity between the *i-th* and *j-th* proposal within the projected hypersphere. Minimizing the above loss function enhances the instance-level similarity among object proposals with identical labels and separates proposals with distinct labels in the projection space.

The network architecture of FSCE, as illustrated in Fig 1, demonstrates how the integration of a contrastive learning branch into the main detection pipeline facilitates enhanced optimization of proposal features, ultimately leading to substantial improvements in detection performance in few-shot scenarios.

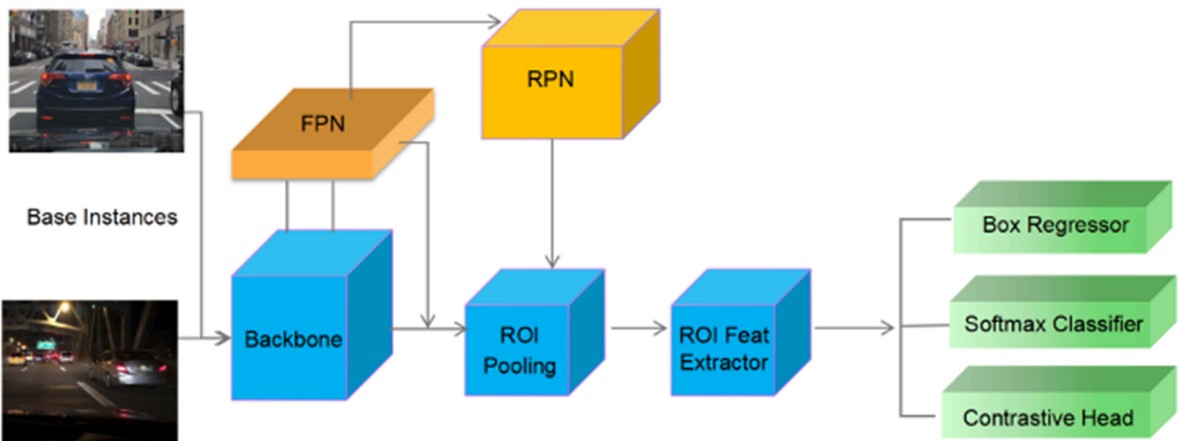

**Fig 1**. **FSCE network structure.** FSCE is based on the Faster R-CNN detection network, including the backbone network (such as ResNet-50), the Region Proposal network (RPN), and the RoI feature extraction module. After the RoI feature extraction, a contrast branch parallel to the classification regression branch is added. The contrast loss optimizes the feature embedding, making the features of similar instances more compact and those of different types more dispersed. The training process first trains the standard Faster R-CNN with rich Base class data (Base Instances), then fine-tuning with new Instances and random balanced data (Novel Instances), freezing the backbone network parameters, and only fine-tuning the RoI feature extractor, classifier, regressor and contrast branch. Train by combining a small number of base class and new class samples. dominating the gradient descent for novel instances in fine-tuning

## 0.2 Improved few-shot object detection based on FSCE

This paper presents an improved FSCE-based framework that introduces two major enhancements to further boost few-shot object detection performance. First, a multi-scale feature enhancement module is incorporated to enrich the representation capacity of the model. By integrating local information and contextual cues alongside the original instance-level features, the model captures discriminative patterns at three different spatial scales. This enables the detection and classification components to access more comprehensive and robust feature representations from limited data, thereby enhancing the learning effectiveness for novel categories.

Second, the conventional Softmax classifier in the detection head is replaced with a cosine Softmax-based classifier. This modification promotes tighter intra-class clustering and reduces feature variance by normalizing feature vectors and leveraging cosine similarity for classification. The adoption of the cosine classifier proves particularly advantageous in few-shot settings, where the model must balance performance between base and novel classes despite data scarcity. It enables more stable generalization and improved accuracy in detecting novel object categories.

The overall architecture of the proposed model, integrating these two enhancements within the FSCE framework, is illustrated in Fig 2. This structure demonstrates how multi-scale feature fusion and cosine-based classification collaboratively contribute to improved few-shot detection performance while preserving the efficiency and end-to-end nature of the original FSCE design.

**0.2.1 Multi-scale feature enhancement.** Some prior studies [18,21] employ fixed-resolution feature pooling strategies, which can result in significant information loss under few-shot conditions due to the limited availability of training images per class. To address this limitation, this paper proposes an approach that integrates local and contextual information alongside traditional instance-level features. These three types of features correspond to distinct spatial regions: the instance-level region, the local region, and the context region, respectively.

The instance-level region typically aligns with the ground truth bounding box, closely enclosing the object of interest. In contrast, the local region refers to the most discriminative sub-region within the instance, which is critical for precise classification and localization. Augmenting the feature representation with local information helps the model focus on the

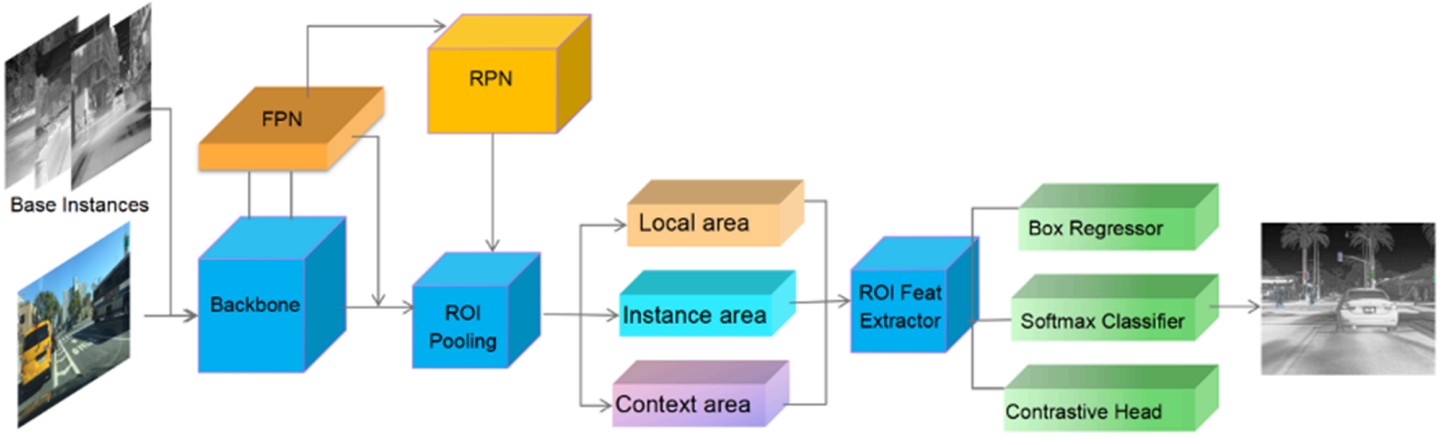

**Fig 2. Overall architecture of the improved FSCE-based few-shot object detection framework.** The improved FSCE follows the original process of "training the base class first and then the new class", with the core optimization focusing on two points: First, it adds a scale feature enhancement operation before the feature extraction stage. On the basis of retaining the original instance-level information, it further integrates local details and context information to improve the learning effect of fest-sample categories. Secondly, replace the border classifier with a design that uses cosine Softmax loss. This approach can effectively balance the learning weights of the base class and the new class, helping to improve the detection accuracy of the target category.

most salient visual cues. Meanwhile, the context region encompasses the background surrounding the object, providing valuable environmental cues that support more accurate recognition in complex or ambiguous scenes.

As illustrated in Fig 3, for each region proposal generated by the Region Proposal Network (RPN), the corresponding local and context regions are defined as rectangular boxes sharing the same center point as the instance-level proposal, but scaled by specific width and height ratios. This region-wise enhancement strategy allows the model to capture complementary information at different spatial scales, thereby improving the robustness and discriminative power of feature representations under few-shot learning constraints.

**0.2.2 Loss function.** In few-shot object detection, the limited availability of training samples for novel categories often results in feature norm discrepancies after network processing, as noted in prior research [25]. These discrepancies adversely affect model training and generalization performance. Moreover, the final classification and regression stages of typical detection models rely on fully connected layers, which are particularly sensitive to data scarcity. The large parameter space associated with fully connected layers poses substantial challenges for effective fine-tuning and optimization in the few-shot regime.

To address these limitations, this paper replaces the conventional Softmax-based classifier with a cosine Softmax classifier in the detection head. Unlike the traditional classifier, which directly maps features to class scores through learned weights, the cosine Softmax formulation normalizes both feature vectors and classifier weights, thereby performing classification based on angular similarity. This design encourages tighter clustering of intra-class features and promotes greater inter-class separability. By reducing intra-class variance and mitigating the impact of norm disparities, the cosine classifier improves the discriminability of feature representations.

The conventional Softmax loss separates features from different categories by maximizing the posterior probability of the true category, that is as shown in Formula 3:

$$L_s = \frac{1}{Z} \sum_{i=1}^{N} -\ln(e^{f_{y_i}} / \sum_{j=1}^{c} e^{f_j}) \tag{3}$$

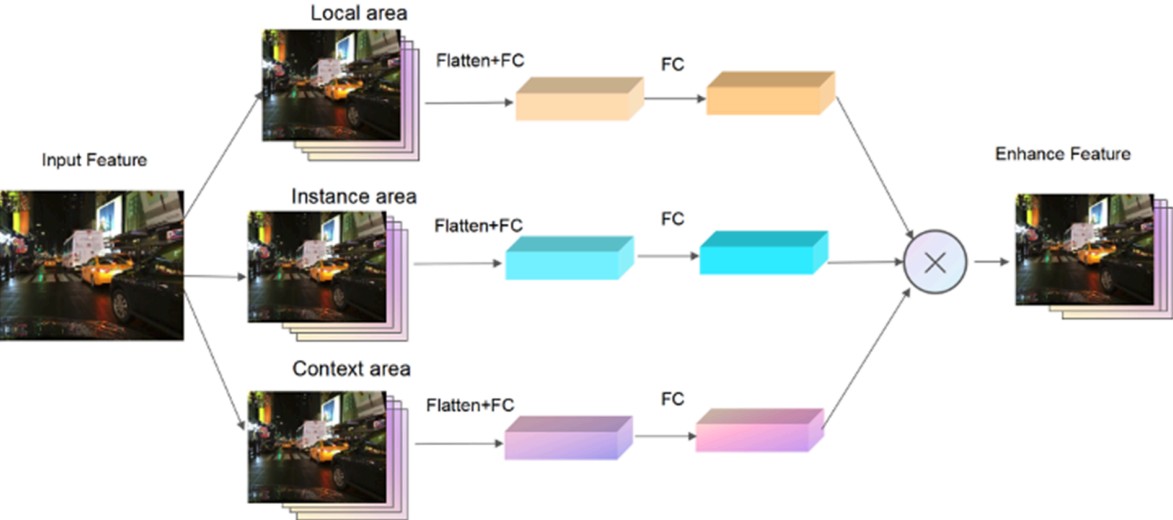

**Fig 3**. **Multi-scale enhanced feature maps.**

In the formula: Z represents the number of training samples, and c represents the number of categories. To conduct effective feature learning, additional constraints are imposed on the Softmax loss through L2 normalization orders $\|w_j\| = 1$. The cosine Softmax loss classifier calculates the classification score based on the cosine similarity between two feature vectors. Therefore, let $\|x\| = \partial$, o is the proportional factor, and thus the posterior probability only depends on the angular cosine. From this, the loss function of the cosine Softmax loss classifier is obtained as Formula 4:

$$L_{\cos} = \frac{1}{N}\sum_{i=1}^{N} -\ln \frac{e^{\partial \cdot \cos\theta_{y_i},i}}{\sum_{j=1}^{C} e^{\partial \cdot \cos\theta_{j,i}}} \tag{4}$$

Because of the fixation $\|x\| = a$, radial variations are eliminated, and the resulting model learns separable features in the angular space. Moreover, $L_{\cos}$ gathers the features of similar samples centered on the classification weight vector, reducing the intra-class variance and improving the detection accuracy for new classes.

In the context of few-shot learning, the cosine Softmax classifier offers a more balanced treatment of base and novel classes, facilitating stable optimization and enhancing the overall accuracy of novel category detection. This modification proves particularly effective in alleviating the overfitting tendencies of conventional classifiers when operating under data-scarce conditions.

## Experiment and results

The experiments were conducted on a system equipped with an Intel Core i7-10700 CPU and an NVIDIA GeForce RTX 3060 GPU, running the Unix Professional Edition operating system. The implementation was based on the PyTorch deep learning framework. Model training was performed using the Stochastic Gradient Descent (SGD) optimizer with an initial learning rate of 0.001. The momentum coefficient was set to 0.937. The training batch size was set to 4, and the total number of training iterations was 30,000. For feature extraction, a ResNet-101 backbone pre-trained on a large-scale dataset was employed to leverage transfer learning and accelerate convergence.

### 0.3 Dataset

This study employs the FLIR and BDD100K [26] datasets to evaluate the detection performance of the proposed improved model. A detailed overview of each dataset is provided below:

The FLIR dataset is a specialized thermal infrared imaging dataset designed for deep learning-based object detection tasks, particularly within the domains of advanced driver-assistance systems (ADAS) and autonomous driving. The dataset comprises a total of 14,452 thermal images, including 10,228 frames extracted from multiple short video sequences and 4,224 frames derived from a continuous 144-second video. These data were collected in diverse real-world environments such as urban streets and highways, encompassing various temporal and weather conditions. The dataset is partitioned into a training set (8,862 images), a validation set (1,366 images), and a video set (4,224 images). Object categories annotated in the dataset include pedestrians, bicycles, cars, dogs, and larger vehicles (e.g., trucks, boats, and trailers). In addition to thermal imagery, the FLIR dataset provides corresponding visible-spectrum images, thereby supporting multimodal analysis and enabling comparative studies between different sensor modalities.

The BDD100K dataset (Berkeley DeepDrive 100K), released by the DeepDrive Lab at the University of California, Berkeley, is one of the largest and most diverse open-source datasets for autonomous driving research. It is designed to enhance the perception, prediction, and decision-making capabilities of autonomous systems in complex driving environments. The dataset captures data across a wide range of conditions, including various times of day (daytime and nighttime), weather scenarios (e.g., sunny, rainy, snowy, foggy), and lighting conditions. It spans multiple geographic locations, including urban, suburban, and rural areas in the United States and parts of Asia. The dataset encompasses diverse road types such as highways, tunnels, and rural roads, and includes complex dynamic elements such as construction

zones and traffic violations. BDD100K is annotated with 10 object categories, including vehicles, pedestrians, traffic signs, and traffic lights, making it highly suitable for evaluating object detection models under real-world, multimodal, and multi-scenario conditions.

## 0.4 Results and discussion

To rigorously assess the performance gains introduced by the proposed improvements, a comparative evaluation is conducted between the enhanced FSCE network and the original FSCE model. This comparison enables a systematic analysis of the impact of the proposed multi-scale feature enhancement and cosine Softmax classification on few-shot object detection performance.

**0.4.1 Analysis of experimental results on FLIR dataset.** In this study, 9 categories from the FLIR dataset were selected for evaluation. Among them, 7 categories were designated as base classes, while the remaining 2 categories—bike, and motor—were defined as novel classes. A two-stage training protocol was adopted. In the first stage, the model was trained on the base class data from the FLIR dataset. Subsequently, few-shot fine-tuning was conducted using a limited number of annotated samples from the novel classes. During both training and evaluation, the N-way K-shot setting was strictly followed, where N denotes the number of novel classes and K represents the number of support instances per class. Specifically, K was set to values of 1, 2, 3, 5, and 10 in different experimental configurations. For consistency and robustness, the Novel Split1 random grouping strategy was employed, and evaluations were performed on test sets corresponding to each category.

In all result tables presented in this study, a "+" symbol denotes performance improvement over the FSCE baseline, while a "−" indicates a decline. The best-performing results across all compared algorithms are highlighted in boldface. As shown in Table 1, the improved model achieves consistent enhancements in mean average precision (mAP) across multiple categories. Notably, the integration of multi-scale feature enhancement and the modified cosine-based loss function leads to the most significant gains in mAP, particularly across 9 of the 10 evaluated categories.

Further, Table 2 presents a detailed comparison of average precision (AP) metrics across different algorithms and evaluation thresholds. Compared to the original FSCE model, the proposed method achieves substantial improvements, registering gains of 2.61, 6.98, 1.96, 2.95, 7.32, 2.76, 1.27, and 5.62 percentage points in AP, AP50, AP75, BAP, BAP50, BAP75, NAP, and NAP50, respectively, across the 10 selected categories.

Based on the above experimental findings, it can be conclusively stated that the proposed enhancements significantly outperform the baseline FSCE algorithm in terms of detection accuracy and generalization capability under few-shot learning conditions.

To more comprehensively evaluate the performance of the proposed model, we also calculated the mAP of different models on the FLIR dataset, that is, the average value of all category AP values, and plotted the corresponding curve graph, as shown in Figs 4–10.

**Table 1**. **The mAP of the second stage in the FLIR dataset under different algorithms.**

| Categories | FSCE | +CBAM | +Loss | +Feature Maps | Ours |
|---|---|---|---|---|---|
| Person | 8.225 | 9.091 | **30.045** | 8.845 | *22.247*(+14.022) |
| Bike | 14.82 | 15.035 | **33.615** | 16.783 | *32.486*(+17.666) |
| Car | 9.091 | 9.091 | *31.1* | 9.091 | **32.66**(+23.569) |
| Motor | 32.1 | 27.712 | *38.023* | 32.123 | **39.587**(+7.487) |
| Bus | 36.14 | 29.869 | 34.09 | *38.326* | **39.922**(+3.758) |
| Truck | **26.684** | *21.94* | 15.856 | 19.83 | 19.118(-7.566) |
| Light | *16.005* | 15.683 | 15.869 | 9.091 | **24.39**(+8.385) |
| Hydrant | 32.755 | 32.271 | *33.444* | 31.333 | **33.774**(+1.019) |
| Sign | 9.091 | 9.091 | **10.95** | 9.091 | *10.729*(+1.638) |

Table 2. Results of the second stage of the FLIR dataset under different algorithms.

| Metrics | FSCE | +CBAM | +Loss | +Feature Maps | Ours |
|---|---|---|---|---|---|
| ap | 10.887 | 10.079 | *12.99* | 10.243 | **13.504**(+2.617) |
| ap50 | 19.517 | 17.969 | *25.391* | 18.528 | **26.503**(+6.986) |
| ap75 | 10.556 | 10.451 | *11.523* | 9.94 | **12.519**(+1.963) |
| bap | 9.019 | 8.724 | *11.492* | 8.506 | **11.973**(+2.954) |
| bap50 | 15.863 | 15.263 | *22.724* | 14.353 | **23.19**(+7.327) |
| bap75 | 9.038 | 9.727 | *10.613* | 8.989 | **11.798**(+2.76) |
| nap | 18.357 | 15.501 | *18.983* | 17.193 | **19.627**(+1.27) |
| nap50 | 34.132 | 28.79 | *36.056* | 35.225 | **39.755**(+5.623) |
| nap75 | **16.63** | 13.35 | 15.164 | 13.745 | *15.401*(-1.229) |

The training dynamics of the proposed few-shot detection framework on the FLIR dataset are illustrated in Figs 4–10, which provide a comprehensive comparison of the baseline FSCE model and its enhanced variants. As shown in Fig 4, the original FSCE model exhibits relatively limited performance improvements during training, serving as a reference for subsequent enhancements. Introducing the CBAM attention module results in the curve displayed in Fig 5, where no observable performance gain is achieved, indicating that CBAM is not well suited to the few-shot detection setting. In contrast, the multi-scale feature enhancement module yields a clear improvement in convergence behavior, as evidenced by

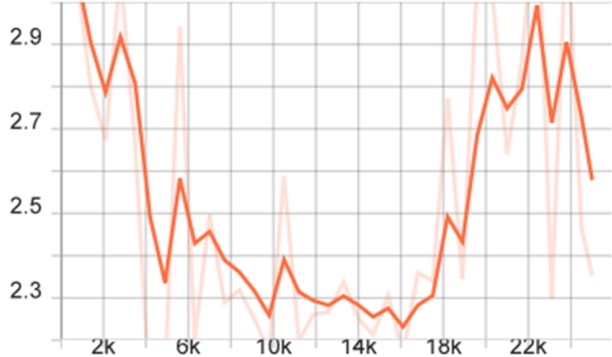

**Fig 4**. mAP Curve graph of the FSCE model on the FLIR dataset.

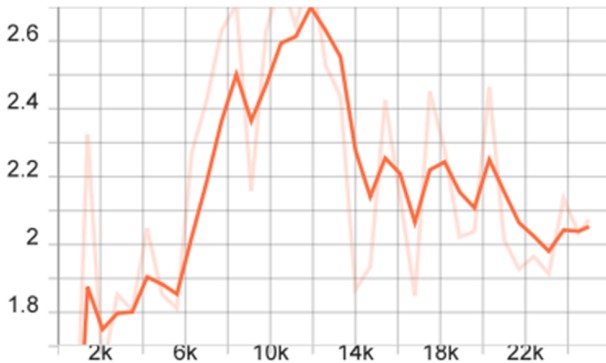

**Fig 5**. mAP curve graph of the FSCE model on the FLIR dataset after adding the CBAM module.

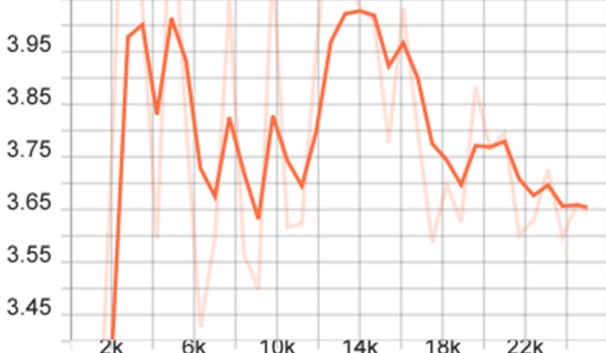

**Fig 6**. mAP curve graph of the FSCE model on the FLIR dataset after adding the multi-scale feature enhancement module.

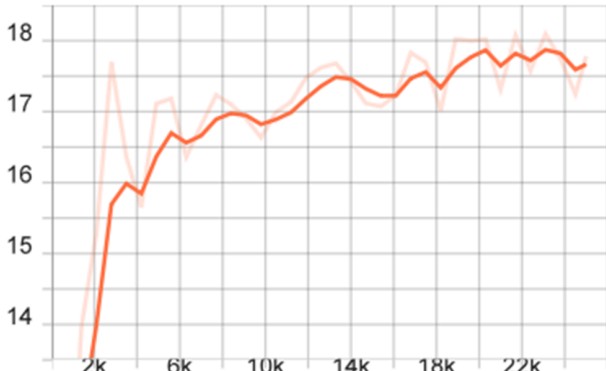

**Fig 7**. mAP curve graph on the FLIR dataset after improving the FSCE model loss function.

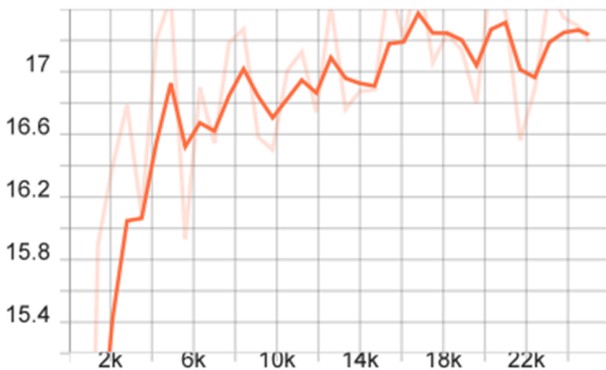

**Fig 8**. mAP curve graph of the improved FSCE model (CBAM, loss function) on the FLIR dataset.

the more stable and higher mAP values shown in Fig 6. A more substantial boost is observed when the standard Softmax classifier is replaced with the cosine Softmax loss; Fig 7 demonstrates enhanced feature separability and improved optimization efficiency. When both CBAM and the cosine-based classifier are incorporated, the training curve in Fig 8 shows that the cosine component continues to offer benefits, although CBAM introduces interference that limits overall

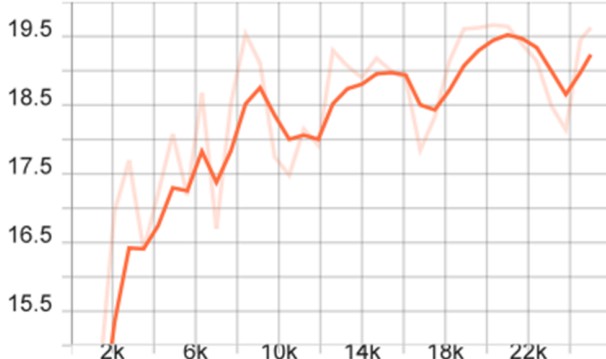

**Fig 9**. mAP curve graph of the improved FSCE model (multi-scale enhancement, loss function) on the FLIR dataset.

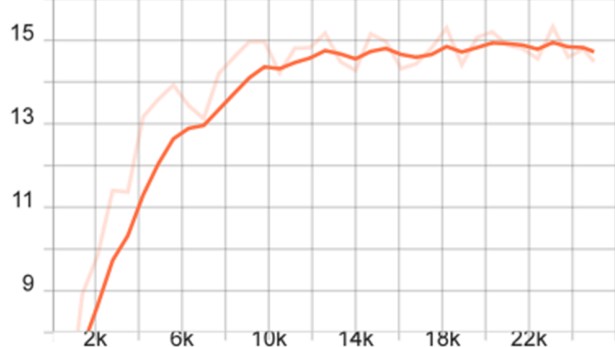

**Fig 10**. mAP curve graph of the improved FSCE model (CBAM, multi-scale feature enhancement, loss function) on the FLIR dataset.

gains. The combination of multi-scale enhancement and cosine Softmax loss results in the most notable improvement, as depicted in Fig 9, reflecting the synergistic effect of enriched multi-scale representations and angular-margin-based classification. Finally, Fig 10 presents the training curve of the full model with all modules enabled, further confirming that the most effective configuration is achieved by integrating multi-scale enhancement with the cosine Softmax loss while excluding CBAM. Together, these curves validate the effectiveness of the proposed improvements in enhancing few-shot detection performance.

Qualitative detection results on the FLIR dataset further demonstrate the robustness of the proposed model under challenging infrared imaging conditions. As shown in Fig 11, the model accurately identifies traffic lights in nighttime scenes, successfully capturing small thermal signatures despite significant background interference and low visibility. This ability to detect fine-grained and distant infrared targets highlights the model's enhanced sensitivity to thermal cues, which is essential for autonomous driving applications in poor lighting environments. Additionally, Fig 12 presents examples of pedestrian detection in urban nighttime settings. The model effectively distinguishes pedestrian shapes from complex thermal backgrounds, suppressing noise while maintaining precise localization. These results confirm that the proposed framework generalizes well to real-world infrared scenes and provides stable detection performance even when visual contrast is low and object features are ambiguous.

**0.4.2 Analysis of experimental results on BDD100K dataset.** This section evaluates the effectiveness of the proposed method through a comparative analysis of experimental results on the BDD100K dataset. The performance of the improved model is benchmarked against the baseline FSCE algorithm and other few-shot detection methods.

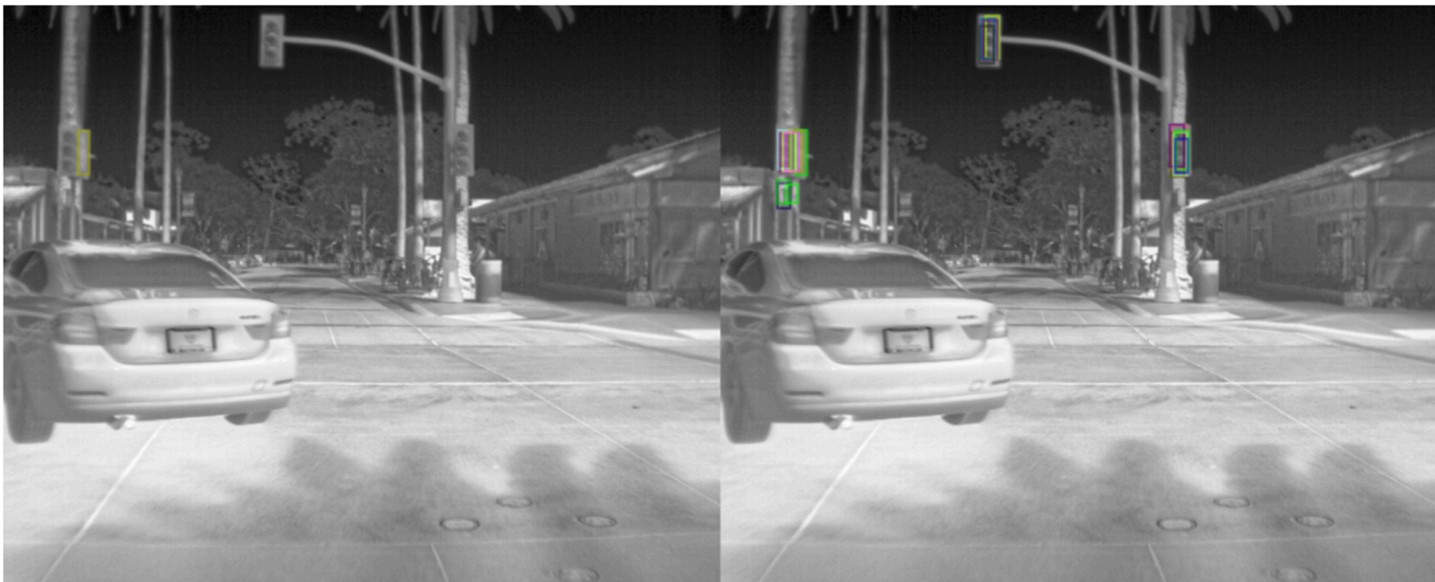

**Fig 11**. **Detection results on the FLIR dataset-traffic light detection.**

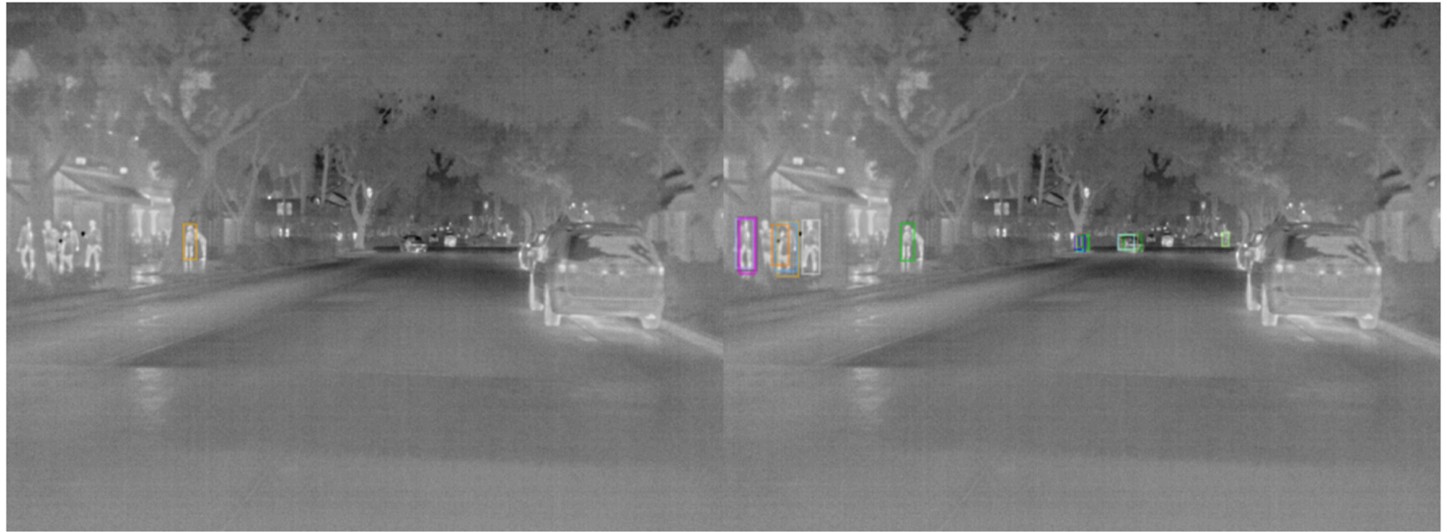

**Fig 12**. **Detection Results on the FLIR Dataset - Pedestrian Detection in Urban Scenes.**

Table 3 reports the mean average precision (mAP) values across nine object categories under various algorithms. A close examination reveals that the incorporation of the multi-scale feature enhancement strategy and the modified cosine similarity-based loss function results in consistent performance improvements across all evaluated categories. These enhancements enable the model to extract more discriminative features and better capture contextual and scale-related information, thereby improving the generalization capability in few-shot scenarios.

Table 4 provides a detailed breakdown of several key evaluation metrics, including average precision (AP, AP50, AP75), bounding box average precision (BAP, BAP50, BAP75), and normalized average precision (NAP, NAP50,

**Table 3**. **The mAP of the second stage in the BDD100K dataset under different algorithms.**

| Categories | FSCE | +CBAM | +Loss | +Feature Maps | Ours |
|---|---|---|---|---|---|
| Car | 9.091 | 9.091 | **29.63** | 9.091 | *22.178*(+13.087) |
| Bus | 19.4 | 18.661 | *23.93* | 18.975 | **25.886**(+6.486) |
| Person | 9.091 | 9.091 | 7.005 | 9.091 | **20.990**(11.899) |
| Bike | 6.3 | 3.138 | *13.025* | 12.317 | **15.288**(+8.988) |
| Truck | 19.742 | 23.907 | 23.907 | 20.307 | **24.513**(+4.771) |
| Motor | 4.459 | **7.643** | *7.115* | 5.527 | 6.618(+2.159) |
| Rider | *8.718* | 4.545 | 6.317 | **10.49** | 4.187(-4.531) |
| Sign | 6.494 | 9.091 | **14.381** | 9.091 | *12.651*(+6.157) |
| Light | 9.091 | 9.091 | *13.141* | 9.091 | **16.480**(+7.389) |

**Table 4**. **Results of the second stage of the BDD100K dataset under different algorithms.**

| Metrics | FSCE | +CBAM | +Loss | +Feature Maps | Ours |
|---|---|---|---|---|---|
| ap | 4.99 | 4.656 | *6.553* | 5.617 | **6.811**(+1.82) |
| ap50 | 9.238 | 8.742 | *13.849* | 10.398 | **14.887**(+5.649) |
| ap75 | 4.679 | 4.699 | **5.208** | *5.206* | 4.738(+0.059) |
| bap | 6.119 | 5.763 | *8.093* | 6.46 | **8.497**(+2.378) |
| bap50 | 10.415 | 6.124 | *16.004* | 10.807 | **17.539**(+7.124) |
| bap75 | 6.449 | 6.124 | *6.66* | **7.09** | 6.356(-0.093) |
| nap | 2.354 | 2.073 | *2.961* | **3.648** | 2.878(+0.524) |
| nap50 | 6.492 | 5.106 | *8.819* | **9.445** | 8.698(+2.206) |
| nap75 | 0.548 | *1.374* | **1.82** | 0.81 | 0.964(+0.416) |

NAP75). The proposed approach demonstrates notable improvements across all these metrics. Specifically, the method achieves gains of 1.821, 5.649, 0.059, 2.378, 7.124, 0.524, 2.206, and 0.416 percentage points in AP, AP50, AP75, BAP, BAP50, NAP, NAP50, and NAP75, respectively, compared to the baseline. These results highlight the effectiveness of the proposed enhancements in addressing the challenges of limited sample sizes and improving detection accuracy for novel categories.

The qualitative results on the BDD100K dataset, shown in Figs 13–15, illustrate the effectiveness of the proposed method across diverse daytime and nighttime driving scenarios. Fig 13 highlights the model's capability to detect small

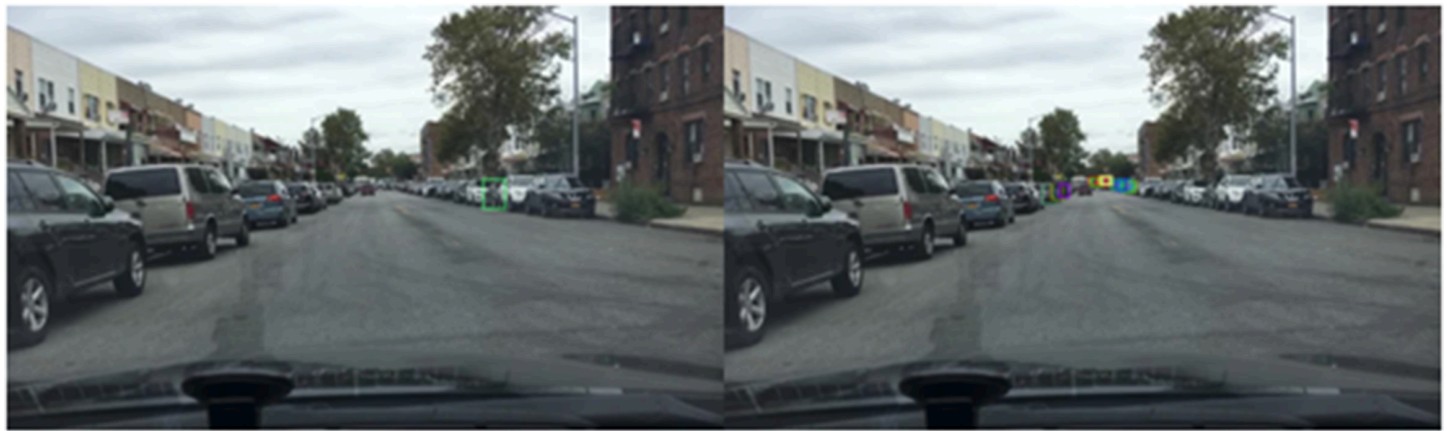

**Fig 13**. **Detection Results on the BDD100K Dataset - Detection of Small and Distant Objects.**

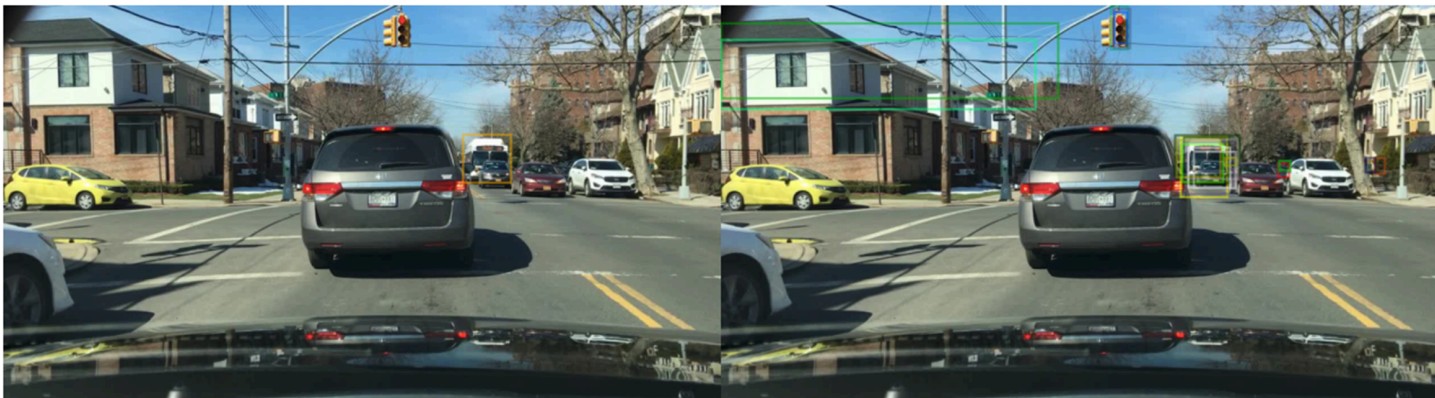

**Fig 14**. Detection Results on the BDD100K Dataset - Detection of Distant Objects and Traffic Signs.

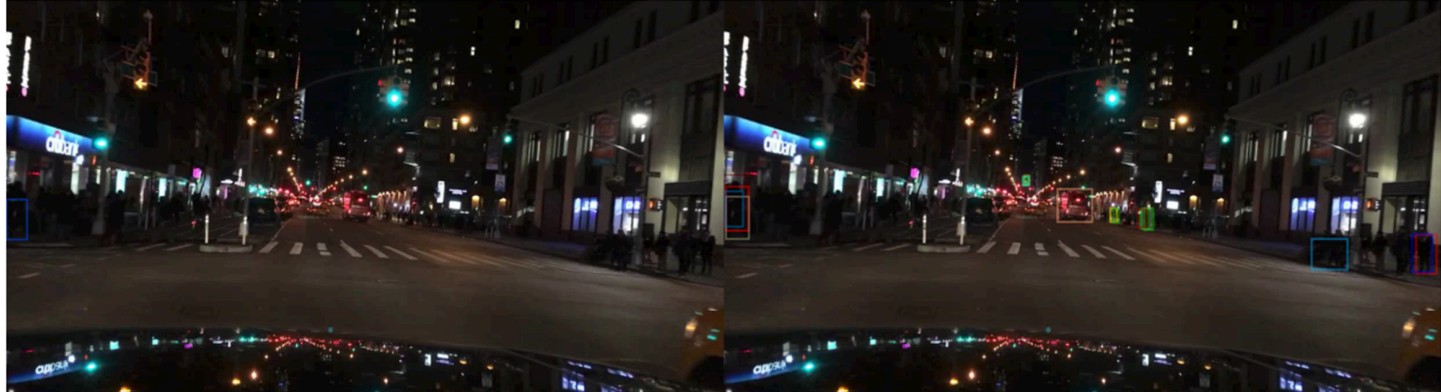

**Fig 15**. Detection Results on the BDD100K Dataset - Detection of Pedestrians and Vehicles at Night.

and distant objects, such as cyclists and compact vehicles, even under difficult perspectives and cluttered surroundings. This demonstrates strong feature sensitivity to scale variations in few-shot settings. In Fig 14, the model maintains high detection accuracy for distant vehicles and traffic signs, successfully distinguishing relevant targets from complex road environments typically encountered in daytime driving. Furthermore, Fig 15 presents detection results in nighttime scenarios, where the model consistently identifies pedestrians and vehicles despite reduced illumination and increased visual noise. Collectively, these qualitative observations validate the generalization ability of the improved FSCE framework and highlight its adaptability to heterogeneous driving conditions, reinforcing its suitability for practical multi-scenario autonomous driving perception.

Overall, the analysis confirms that the integration of multi-scale feature representations and cosine-based classification substantially enhances few-shot object detection performance, validating the practicality and robustness of the proposed method.

**0.4.3 Ablation study.** To evaluate the effectiveness of each proposed module, a series of ablation experiments were conducted. The results, presented in Tables 1 to 4, assess both the individual and combined contributions of the modules to the overall model performance. These results demonstrate that each module contributes to improving detection accuracy, with the most significant impact observed from the cosine loss function, followed by the multi-scale feature enhancement module. Specifically, incorporating the multi-scale feature enhancement module into the baseline FSCE model

yields a modest improvement in detection accuracy, thereby confirming its utility. Further enhancement is achieved by replacing the original loss function with a cosine-based loss, which leads to a more substantial performance gain. These findings validate the effectiveness of the proposed modules in enhancing the baseline model.

To more intuitively demonstrate the performance of the proposed model, qualitative results were visualized using sample detection outputs on two benchmark datasets, as shown in Figs 4–15. Figs 4–10 illustrate the different training performances of different algorithms. On the FLIR dataset, the model effectively identifies key targets such as pedestrians and vehicles in nighttime scenes. Leveraging the advantages of infrared imaging, it accurately captures the thermal radiation characteristics of targets even under low-light conditions, while effectively suppressing background noise, thereby exhibiting strong robustness. For instance, in complex nighttime street scenes, the model is capable of precisely localizing distant pedestrians while avoiding false detections of irrelevant objects such as streetlights.

On the BDD100K dataset, the model successfully detects a diverse set of targets in daily driving scenarios—including vehicles, pedestrians, and traffic signs—under varying lighting conditions. The integration of the attention mechanism and the multi-scale feature enhancement module enables the model to better manage complex backgrounds and multi-scale targets, significantly improving both detection accuracy and recall. The visualization results confirm that the model consistently delivers high-quality detection outcomes, even in few-shot scenarios, thus demonstrating strong generalization capability and practical applicability across diverse environments.

However, comparative experiments revealed that the inclusion of the CBAM module did not lead to performance gains; rather, it resulted in a degradation of detection accuracy. This suggests that, under the current experimental setup and data conditions, the CBAM module failed to enhance feature representation and attention mechanisms effectively. Moreover, the added computational complexity may have introduced overhead that adversely affected the model's performance. The proposed model exhibited favorable results in few-shot vehicle detection tasks and achieved competitive performance on both the FLIR and BDD100K datasets. These findings further validate the effectiveness and practicality of the proposed approach.

## Conclusion

This paper proposes a novel few-shot object detection model based on the FSCE framework, addressing the persistent challenge of data scarcity in object detection tasks. The proposed approach incorporates two key enhancements. First, a multi-scale feature enhancement strategy is employed to effectively handle target objects of varying sizes, thereby improving the model's robustness across diverse spatial contexts. Second, a cosine similarity-based classification loss is introduced to replace the conventional Softmax loss. This modification facilitates tighter intra-class feature aggregation and reduced intra-class variance, thereby significantly enhancing classification accuracy under limited data conditions.

Extensive experiments conducted on the FLIR and BDD100K datasets demonstrate that the proposed method consistently outperforms baseline models and several state-of-the-art few-shot object detection algorithms, particularly in scenarios with limited labeled samples. These results validate the efficacy of the proposed enhancements in improving both the accuracy and generalization capability of the model. Nevertheless, few-shot object detection remains an open research problem. Currently, many approaches rely heavily on two-stage detection frameworks, which, although effective in refining proposal generation, often introduce additional computational complexity and impose substantial memory demands. Future work will explore lightweight architectures and single-stage detection frameworks that can maintain high accuracy while improving efficiency and scalability, thereby advancing the practical deployment of few-shot detection systems in resource-constrained environments. In the future, the work will be advanced around four aspects: First, deploy edge devices, complete hardware selection and debugging, and build a hardware environment; The second is to optimize the algorithm and achieve lightweighting through pruning and other means; The third is to enhance the model's accuracy, iterate on existing methods and try new solutions; The fourth is to introduce large language models to replace traditional neural networks and innovate the technology.

## Author contributions

**Conceptualization:** Yalei Dong, Fengchen Wei.

**Data curation:** Yalei Dong, Jing Xiao.

**Formal analysis:** Yalei Dong.

**Methodology:** Yalei Dong.

**Project administration:** Fengchen Wei.

**Resources:** Yalei Dong.

**Software:** Yalei Dong, Jing Xiao.

**Supervision:** Fengchen Wei.

**Validation:** Jing Xiao.

**Visualization:** Jing Xiao.

**Writing – original draft:** Yalei Dong.

**Writing – review & editing:** Fengchen Wei.

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
