## [Decision Letter · Decision Letter 0]

29 Aug 2025

PONE-D-25-27923A Novel Few-Shot Object Detection Framework for Multi-Scene Driving Based on Contrastive Proposal EncodingPLOS ONE

Dear Dr. Wei,

Thank you for submitting your manuscript to PLOS ONE. After careful consideration, we feel that it has merit but does not fully meet PLOS ONE’s publication criteria as it currently stands. Therefore, we invite you to submit a revised version of the manuscript that addresses the points raised during the review process.

We look forward to receiving your revised manuscript.

Kind regards,

Jinhao Liang

Academic Editor

PLOS ONE

Journal Requirements:

Additional Editor Comments:

Please revise the paper throughly according to the reviewers' comments. Furthermore, the background should highlight the rapid development of autonomous vehicles, thereby enhancing the research value of this work. The recent work "Enhancing High-Speed Cruising Performance of Autonomous Vehicles Through Integrated Deep Reinforcement Learning Framework, IEEE Transactions on Intelligent Transportation Systems, vol. 26, no. 1, pp. 835-848, Jan. 2025" can be referred.

Reviewers' comments:

Reviewer's Responses to Questions

**Comments to the Author**

1. Is the manuscript technically sound, and do the data support the conclusions?

Reviewer #1: Partly

Reviewer #2: Yes

2. Has the statistical analysis been performed appropriately and rigorously?

Reviewer #1: Yes

Reviewer #2: N/A

3. Have the authors made all data underlying the findings in their manuscript fully available?

Reviewer #1: Yes

Reviewer #2: Yes

4. Is the manuscript presented in an intelligible fashion and written in standard English?

Reviewer #1: Yes

Reviewer #2: Yes

5. Review Comments to the Author

Reviewer #1: This paper PONE-D-25-27923 proposes enhancements to the FSCE framework for few-shot object detection in driving environments, introducing a multi-scale feature enhancement module and cosine Softmax classifier. While the work addresses a relevant problem, several significant issues limit its contribution.

- Multi-scale feature extraction is a well-established technique in computer vision. The proposed approach of using instance, local, and context regions is incremental and lacks rationale behind the specific scale ratio selections.

- The cosine Softmax classifier is well-established in face recognition. The author should clearly articulate their novel contribution beyond standard implementation and provide mathematical formulation for their cosine Softmax modification.

- You must compare against recent few-shot detection methods (DeFRCN 2021, Meta-DETR 2022, etc.) - comparing only to FSCE is insufficient

- Add statistical validation: report error bars, confidence intervals, and significance tests for all results.

- Include computational analysis (FPS, memory usage, parameters) since you claim real-time applicability.

- Conduct cross-dataset experiments to demonstrate generalization capability.

- Clarify your train/test split methodology and cross-validation approach

- Some figures have low resolution and small fonts. It is recommended to standardize and improve the quality of the figures to enhance readability and comprehension for readers. Figure 2 needs more detail about your multi-scale fusion mechanism. Improve Figures 4-8: add clearer bounding box annotations and confidence scores.

- Make Table formatting consistent and define all metrics clearly.

- Although some references are more or less recent, there are many (38%) are old references. Please, update the references with latest state-of-the-art research.

- Please, replace the arXiv references with their respective final published version.

- Authors should remove all typos like Line 227 "GTX 3060" to "RTX 3060".

- Reference [17] appears to be a duplicate of [15].

Reviewer #2: The manuscript entitled “A Novel Few-Shot Object Detection Framework for Multi-Scene Driving Based on Contrastive Proposal Encoding”. Authors enhanced a few-shot object detection algorithm depended on Contrastive Proposal Encoding in order to improve both the accuracy and generalization capability of the model. They employed two types of datasets to evaluate the detection performance. However, the manuscript has several critical weaknesses.

6. PLOS authors have the option to publish the peer review history of their article (what does this mean?). If published, this will include your full peer review and any attached files.

Reviewer #1: No

Reviewer #2: **Yes: **Auhood Al-Hossenat

---

## [Author Response · Author response to Decision Letter 1]

21 Oct 2025

Dear Editors and Reviewers:

Thank you for your constructive comments, which helped us improve the quality of our manuscript.

We have taken these comments into account and incorporated your valuable suggestions in the revised manuscript,

with particular emphasis on the key contributions of this article. For detailed revisions, please refer to our "Response to Reviewers" and the revised "Manuscript”.

---

## [Decision Letter · Decision Letter 1]

10 Nov 2025

A Novel Few-Shot Object Detection Framework for Multi-Scene Driving Based on Contrastive Proposal Encoding

PONE-D-25-27923R1

Dear Dr. Wei,

We’re pleased to inform you that your manuscript has been judged scientifically suitable for publication and will be formally accepted for publication once it meets all outstanding technical requirements.

Kind regards,

Jinhao Liang

Academic Editor

PLOS ONE

Additional Editor Comments (optional):

Reviewers' comments:

Reviewer's Responses to Questions

**Comments to the Author**

1. If the authors have adequately addressed your comments raised in a previous round of review and you feel that this manuscript is now acceptable for publication, you may indicate that here to bypass the “Comments to the Author” section, enter your conflict of interest statement in the “Confidential to Editor” section, and submit your "Accept" recommendation.

Reviewer #1: All comments have been addressed

Reviewer #2: All comments have been addressed

2. Is the manuscript technically sound, and do the data support the conclusions?

Reviewer #1: Yes

Reviewer #2: Yes

3. Has the statistical analysis been performed appropriately and rigorously?

Reviewer #1: Yes

Reviewer #2: Yes

4. Have the authors made all data underlying the findings in their manuscript fully available?

Reviewer #1: Yes

Reviewer #2: Yes

5. Is the manuscript presented in an intelligible fashion and written in standard English?

Reviewer #1: Yes

Reviewer #2: Yes

6. Review Comments to the Author

Reviewer #1: (No Response)

Reviewer #2: (No Response)

7. PLOS authors have the option to publish the peer review history of their article (what does this mean?). If published, this will include your full peer review and any attached files.

Reviewer #1: No

Reviewer #2: No

---

## [Editor Report · Acceptance letter]

PONE-D-25-27923R1

PLOS One

Dear Dr. Wei,

I'm pleased to inform you that your manuscript has been deemed suitable for publication in PLOS One. Congratulations! Your manuscript is now being handed over to our production team.

Kind regards,

on behalf of

Dr. Jinhao Liang

Academic Editor

PLOS One